# Biochar and Compost Application either Alone or in Combination Affects Vegetable Yield in a Volcanic Mediterranean Soil

Giuseppina Iacomino [1] , Tushar C. Sarker [2], Francesca Ippolito [1], Giuliano Bonanomi [1,3], Francesco Vinale [4] , Alessia Staropoli [1,5] and Mohamed Idbella [1,6,*]

1    Department of Agricultural Science, University of Naples Federico II, Via Università 100, 80055 Naples, Italy
2    School of Environmental and Resource Sciences, Zhejiang A&F University, Jinhua 311300, China
3    Task Force on Microbiome Studies, University of Naples Federico II, 80055 Naples, Italy
4    Department of Veterinary Medicine and Animal Productions, University of Naples Federico II, 80137 Naples, Italy
5    Institute for Sustainable Plant Protection, National Research Council, 80055 Naples, Italy
6    Laboratory of Biosciences, Faculty of Sciences and Techniques, Hassan II University, Casablanca 28806, Morocco
*    Correspondence: mohamed.idbella@usmba.ac.ma; Tel.: +39-(334)-980-8982

**Abstract:** The aim of this work was to compare the application of biochar, compost, and their mixtures on soil fertility and crop yields using a volcanic Mediterranean soil. For this reason, three types of organic amendments (OAs) were selected: compost1, made from olive mill waste and orchard pruning residues; compost2, made from olive mill waste, animal manure and wool residues; and biochar made from beech wood pyrolyzed at 550 °C. When selected, the OAs were characterized chemically for organic carbon (OC), total N, pH, electric conductivity (EC) and the bulk fraction of organic matter using $^{13}C$ CPMAS NMR spectroscopy. In addition, soil chemistry was analysed at the end of each year for the following parameters: pH, OC, total N, $CaCO_3$, $P_2O_5$, $NH_4$, FDA and EC. Results showed that biochar had the highest OC and the lowest N and EC compared to both composts. Moreover, $^{13}C$ CPMAS NMR showed that biochar had the lowest content of O-alkyl, methoxyl- and alkyl-C and the highest content of aromatic-C. On the other hand, compost2 and compost2+biochar mixture reduced Aubergine yield by −60% and −40%, respectively, and tomato yield by −50% and −100%, respectively. Nevertheless, a significant increase in onion and rape yields were observed when compost1, compost1+biochar and compost2 were applied, while biochar and compost2+biochar significantly decreased the yield of these crops. Overall, our results highlight that the effect of OAs on crops yield is largely variable and influenced by the interaction with soil chemistry.

**Keywords:** $^{13}C$ CPMAS NMR; biochar; compost; organic amendment; organic carbon

## 1. Introduction

Agricultural productivity declines due to the decrease in soil organic matter (SOM). Soil nutrients depletion and human-induced climate change are important factors threatening the sustainability of agricultural production [1]. Since the "green revolution", inorganic fertilizers have played a key role in increasing agricultural production and maintaining productivity over the past half century [2]. However, the use of these chemical fertilizers alone is not a sustainable solution for improving soil fertility and maintaining crop yield. Rather, it is widely recognized that overuse of chemical fertilizers, particularly nitrogen, can lead to soil degradation and cause other environmental problems as SOM is mineralized more rapidly, which in turn causes a sharp decline in soil carbon stocks [3].

Maintaining adequate levels of SOM and ensuring effective biological nutrient cycling are essential to the success of soil conservation, management, and agricultural productivity



strategies [4]. These practices include the use of organic and inorganic fertilizers in conjunction with knowledge of how to adapt these practices to local conditions, with the goal of optimizing the agronomic performance of applied nutrients and thus crop productivity [4]. However, the naturally rapid mineralization of SOM is considered an important limitation to the practical application of organic fertilizers. Therefore, a new approach is needed to help increase yields, minimize negative impacts, improve sustainability, and be accessible to both subsistence and commercial producers [5]. Thus, research on biochar-compost mixtures as soil amendments has made significant progress and provided important insights into agronomic benefits, carbon sequestration, greenhouse gas emissions, soil quality, and fertility improvement [6].

Biochar is a solid material produced by the thermochemical conversion of biomass in an oxygen-deficient environment [7]. This production technology is robust, simple, and suitable for many regions of the world [8]. One of the most commonly used biochar is wood based, but this amendment can result in nitrogen immobilization that reduces crop yield in the short term [9]. However, this immobilized nitrogen can be slowly released to plants for uptake, resulting in significantly higher long-term benefits [10]. Recently, the use of biochar in soil has emerged as an approach to sequester carbon, reduce greenhouse gas emissions, and improve soil quality [11]. Studies have shown that the use of biochar can significantly improve SOM [12], water holding capacity [13], soil aeration, soil base saturation, nutrient storage, and availability, and reduce fertilizer requirements and nutrient leaching [6,14]. In addition, biochar has been shown to stimulate the soil microbial community, increase microbial biomass and activity [15], and improve plant growth and yield [11]. However, the addition of pure biochar to soil does not necessarily improve soil quality and crop yields [16]. For this reason, the application of biochar in combination with other organic amendments, such as compost, has recently received attention due to its promising results in both pot and field trials [17,18]. Evidence suggests that a mixture of biochar and fresh organic matter is a powerful combination to explore the potential beneficial effects of such organic amendments on higher plants [19]. In general, various composted materials provide a sustainable source of available nutrients that could enhance plant growth by improving soil physicochemical and microbiological properties [19–21]. Liu et al. [19] demonstrated that the combination of compost and biochar had a positive synergistic effect on soil nutrient content and water holding capacity under field conditions. In addition, this combination was found to be more suitable as it reduced chemical fertilizer use, stabilized soil structure, and improved soil nutrient content and water retention capacity [20]. Moreover, the same studies highlighted that the mixture of compost and biochar can enhance the compost properties, resulting higher added value and much better carbon sequestration potential due to the long-term stability of biochar [20,21].

In general, mixing biochar with organic amendments has been shown to have great potential for improving agricultural productivity [22]. However, literature studies have also shown that these positive effects may depend on various factors such as compost and biochar quality, application rate, soil type, and climatic conditions [6,11–16]. Several studies examine the effects of compost and biochar on crop yields across a range of soil types and climatic conditions [23]. However, far fewer studies have focused on the effects of biochar-compost mixtures, with most of the available studies conducted in tropical and temperate soils, and no studies in volcanic soils in Mediterranean climates. To fill this gap, we conducted a two-year field trial with 15 cropping cycles, including aubergine, fennel, lettuce, onion, pumpkin, rape, and tomato, to evaluate the effects of biochar, compost, and compost-biochar mixtures on crop yield and soil chemistry i.e., pH, organic C, total N, total $CaCO_3$, $P_2O_5$, and electrical conductivity, and their relationship with crop yield were studied. Our specific aims were:

i. To evaluate the effect of biochar and compost alone or in a mixture on crop yield in a volcanic Mediterranean soil.

ii. To assess how biochar and the composts application changes the soil chemical compositions and the activity of enzymes.

## 2. Materials and Methods

### 2.1. Study Site Description and Soil Chemistry

The study site is a productive area dedicated to vegetable cultivation in the open field and in greenhouses located at the Napoli district (southern Italy) on the slopes of Mount Vesuvius. The study area has a Mediterranean climate with a mean annual temperature of 16.2 °C and mean monthly temperatures ranging from 25.9 °C in August to 9.1 °C in January. Total rainfall is high 929 mm per year, with most precipitation in winter (290 mm), spring (200 mm), and fall (348 mm), and a large dry season in summer (89 mm).

The experiment was conducted in the research station of the Gussone Park of the Royal Palace of Portici (40°48′40.3″ N, 14°20′33.8″ E; 75 m a.s.l.), located at the foot of the southwestern slopes of Vesuvius, overlooking the sea of the Gulf of Naples. The experimental station consisted of Andisols with the following characteristics: sand 44.0%, silt 33.0%, clay 23.0%, pH 8.02, organic carbon 22.13 g/kg, total N 1.64 g/kg, C/N 19.07, total $CaCO_3$ 124 g/kg, available phosphorus ($P_2O_5$) 27.3 mg/kg, exchangeable potassium 0.14 meq/100 g, exchangeable magnesium 4.11 meq/100 g, exchangeable calcium 13.6 meq/100 g, exchangeable sodium 0.09 meq/100 g, EC 0.326 dS/m.

### 2.2. Organic Amendments

Three types of organic amendments were selected: (1) compost1, from olive mill waste and orchard pruning residues; (2) compost2, from olive mill waste, animal manure, and wool residues; (3) biochar, produced from beech wood (*Fagus sylvatica*), pyrolyzed at 550 °C. Composting mixtures were prepared by mixing either dried or fresh olive mill waste with different organic materials used as bulking agents and N-sources, such as animal manure, orchard pruning residues, and wool residues. In a pilot plant, composts were prepared using the turning-pile system in trapezoidal piles (2 m long, 1 m wide at the base, and 1 m high). The piles were turned every two weeks during the bio-oxidative phase (approximately 16–20 weeks, from June to October). The mixtures were matured over a two-month period. Temperature and moisture were used as monitoring parameters to track the progress of composting. Water was added during turning to maintain moisture content in the range of 40–60%. On the other hand, biochar of beech wood (*Fagus sylvatica*) was made by slow pyrolysis at 550 °C, and was used as stable carbon material, to improve soil physical properties.

All organic materials were dried, grounded, and sieved with a mesh size of 2 mm and chemically analysed for total C and N content by flash combustion of micro-samples (5 mg of the sample) in an Elemental Analyzer (NA 1500 Fison 1108 Elemental Analyzer, Thermo Fisher Scientific, Waltham, MA, USA). The pH and EC values of the organic amendment were measured using a pH meter and a conductometer, respectively. All organic materials were also analysed by solid-state [13]C CPMAS NMR to obtain detailed molecular composition. The [13]C CPMAS NMR spectra were recorded using a Bruker AV −300 NMR spectrometer (Bruker Instrumental Inc., Billerica, MA, USA) equipped with a 4 mm diameter Magic Angle Spinning (MAS) probe. Briefly, we used the following calibrated acquisition characteristics: 2 s cycle time; 1H power for CP 92.16 W: 1H 90° pulse 2.85 μs; 13C power for CP 1504 W; 1 ms contact time; 20 ms acquisition time; 2000 scans [24]. The spectra were divided into selected regions corresponding to classes of organic carbon bonds [25]. Specifically, the following seven chemical regions representative of the major organic carbon types were considered: 0–45 ppm = alkyl + alpha-amino-C; 46–60 ppm = methoxyl and N-alkyl-C; 61–90 ppm = O-alkyl-C; 91–110 ppm = di-O-alkyl-C; 111–140 ppm = H and C-substituted aromatic C; 141–160 ppm = O-substituted aromatic C; and 161–190 ppm = carbonyl-C. The relative abundance for each selected region was calculated by integrating MestreNova 6.2 software (Mestre-lab Research 2010) and expressed as a percentage of the total spectral area.

### 2.3. Experiment Set-Up and Crop Yields Assessment

At the research station, an area of approximately 2000 m$^2$ was selected and divided into 18 plots to obtain a randomized block design with three field replicates for each of the

treatments plus the untreated control. The plots were 18 m long and 8 m wide and contained six treatments: untreated control; compost1; compost2; biochar; compost1+biochar; and compost2+biochar.

Compost1 and compost2 were applied at a rate of 20 Mg.ha$^{-1}$, while biochar was applied at a rate of 30 Mg.ha$^{-1}$. Compost and biochar were applied once at the beginning of the experiment and incorporated by rototilling. The application rate was based on previous experience with a similar cropping system [26]. For biochar, previous work has shown that higher dosages of up to 80 Mg.ha$^{-1}$ can be more effective in promoting plant growth [27]. However, the application rate > 30 Mg.ha$^{-1}$ would not be economically feasible at current biochar prices in Italy.

Seven vegetable varieties were tested: Aubergine (*Solanum melongena*), Fennel (*Foeniculum vulgare*), Lettuce (*Lactuca sativa*), Onion (*Allium cepa*), Rape (*Cucurbita pepo*), and Tomato (*Lycopersicon esculentum*). Eight crop cycles were conducted in the first year (i.e., three lettuce cycles, one for Aubergine, Fennel, Onion, Rape, and Tomato), and another seven cycles were conducted in the second year (i.e., one for Aubergine, Fennel, Lettuce, Onion, Pumpkin, Rape, and Tomato). Total crop yields were recorded at the end of each cycle to quantify the amount of commercial production in all experimental plots.

### 2.4. Soil Sampling and Analyses

At the end of the first and second year of the experiment, three samples of topsoil (0–30 cm) were collected from each plot according to the W scheme. The soil samples were packed in polyethylene bags, transported to the laboratory, sieved (<2 mm), and air-dried at room temperature for the chemical analyses.

Soil chemical parameters were determined using standard methods [28]. Briefly, electrical conductivity (EC) and pH were determined in soil-water suspensions at a ratio of 1:5 and 1:2.5 using a BASIC 30, CRISON pH meter and a BASIC 30, CRISON conductometer, respectively. Available phosphate was determined by bicarbonate extraction, and organic carbon (C org) content was quantified by the chromic acid titration method [29]. Total N (Nitrogen) content was determined by flash combustion using a CNS Elemental Analyzer (Thermo FlashEA 1112). Ammonium in soil was analysed for NH4 using a DR 3900 spectrophotometer (Hach, Loveland, CO, USA) with the manufacturer's kit LCK 303 (assay range 2–47 mg/L). Finally, soil enzymatic activity was determined using the fluorescein diacetate method (FDA), which measures the amount of protease, lipase, and nonspecific esterase associated with organic matter turnover [30].

### 2.5. Data Analyses

Two-way analysis ANOVA was used to evaluate the effects of the year and experimental treatments on crop yield. With respect to soil chemistry, one-way ANOVA was used to evaluate the effects of the experimental treatments on soil chemical and enzymatic parameters. Significance was evaluated at $p < 0.05$ and <0.01 and statistical analyses were performed using STATISTICA 10 (StatSoft). Moreover, Principal Component Analysis (PCA), a multivariate approach, was performed to evaluate the treatments variability based on chemical and biochemical parameters. The length of each eigenvector is proportional to the variance in the data for that element. The angle between eigenvectors represents correlations among different elements.

## 3. Results

### 3.1. Organic Amendment Chemistry

Wood biochar and the composts showed notable differences in chemical compositions, in terms of elements content and molecular compositions as quantified by $^{13}$C CPMAS NMR spectra. Wood biochar had exceptionally high C/N ratio of 437.55 due to the low N content combined with the very high C concentration. On the other hand, compost types showed lower C/N ratio, specially compost2 had lower C/N ratio (12.54) compared to compost1, C/N ratio 30.64 (Table 1). Electrical conductivity has shown a large variation ranging from

0.13 for wood biochar to 0.96 and 1.18 mS/cm for compost1 and compost2, respectively. The pH was neutral for the two tested composts and basic for wood biochar (Table 1).

**Table 1.** Content of organic carbon, total nitrogen, pH, electric conductivity, C/N ratio and $^{13}$C CPMAS NMR data of the wood biochar and the two-compost type. Different letters within each column indicate significant difference ($p < 0.05$).

| Parameters | Biochar | Compost1 | Compost2 |
|---|---|---|---|
| **Chemical** | | | |
| C % | 87.51 a | 38.30 b | 36.71 b |
| N % | 0.20 c | 1.25 b | 3.18 a |
| C/N ratio | 437.55 a | 30.64 b | 12.54 c |
| pH | 9.28 a | 6.89 b | 6.11 b |
| EC mS/cm | 0.13 b | 0.96 a | 1.18 a |
| **$^{13}$C CPMAS NMR** | | | |
| Carbonyl-C (161–190 ppm) | 4.08 | 7.06 | 8.34 |
| O-subst. aromatic C (141–160 ppm) | 5.23 | 5.75 | 4.00 |
| H-C subst. aromatic C (111–140 ppm) | 65.38 | 12.31 | 6.93 |
| di-O-alkyl C (91–110 ppm) | 5.98 | 11.32 | 6.00 |
| O-alkyl C (61–90 ppm) | 5.29 | 34.54 | 32.59 |
| Methoxyl C (46–60 ppm) | 4.49 | 11.63 | 15.32 |
| Alkyl C (0–45 ppm) | 9.55 | 17.39 | 26.82 |

Considering C bond types derived from NMR spectrum, the alkyl-C (0–45 ppm), and the methoxyl (46–60 ppm) fractions showed high peaks in compost2 followed by compost1, while less abundant in wood biochar (Table 1). The O-alkyl-C (61–90 ppm) region that is associated with polysaccharides and sugars was especially high in both composts and very low in wood biochar. However, the di-O-alkyl-C (91–110 ppm) fraction was equally low in both biochar and compost2 and relatively higher in compost1. The H- and C-substituted aromatic C (111–140 ppm) fraction was very abundant in wood biochar. The O-substituted aromatic C fraction (141–160 ppm) was not abundant in the studied material. Finally, the carbonyl C (161–190 ppm) fraction was slightly high in both composts and less abundant in wood biochar (Table 1).

*3.2. Soil Chemistry*

The chemical compositions of the soil varied significantly between the first and second year of organic treatments (Table 2). Soil pH showed no significant differences between treatments in the first year and decreased significantly after the second year in all treatments. Enzymatic activity (FDA) also decreased significantly in the second year in all treatments. Nevertheless, organic carbon (OC)was significantly higher in biochar alone and in biochar+compost2 treatment in the second year compared to other treatments (Table 2). In addition, total nitrogen significantly increased in the second year compared to the first year for most of the treatments except for compost2, biochar+compost1, and biochar+compost2, where it significantly decreased in the second year. On the other hand, $NH_4$, increased significantly only in compost1 for the first and second year, while EC decreased significantly only in the biochar+compost2 mixture for the two years. $P_2O_5$, however, showed no significant difference between the first and second year in all treatments (Table 2).

**Table 2.** Summary of the Factorial ANOVA testing for main and interactive effects of different treatments on the total biomass of several crops type during a two-years experiment.

| | Sum Squares | Degree of Freedom | Mean Squares | F-Value | *p*-Value |
|---|---|---|---|---|---|
| Treatment | 59,864 | 5 | 11,972 | 980.0283 | 0.2377 |
| Crop Type | 2,331,863 | 5 | 466,372 | 1.3624 | 0.0000 * |
| Year | 2309 | 1 | 2309 | 53.0697 | 0.6085 |

**Table 2.** *Cont.*

|  | Sum Squares | Degree of Freedom | Mean Squares | F-Value | *p*-Value |
|---|---|---|---|---|---|
| Treatment × Crop | 277,540 | 25 | 11,101 | 0.2628 | 0.1810 |
| Treatment × Year | 39,702 | 5 | 7940 | 1.2633 | 0.4787 |
| Crop × Year | 106,372 | 5 | 21,274 | 0.9036 | 0.0353 * |
| Treatment × Crop × Year | 144,659 | 25 | 5786 | 2.4208 | 0.8959 |

(* The mean difference is significant at the 0.05).

### 3.3. Vegetable Yields

Total crop yield showed great variability within each treatment over the two years (Figure 1). Crop yield was significantly affected by crops type and the interaction between crop type and experimental year (Table 3). All five different treatments either increased or decreased crop yield compared to control during the transition from the first to the second year (Figure 2). The compost2 and compost2+biochar combination decreases annual yield from −10% to −60% and from −17% to −40%, respectively, for Aubergine, and from 21% to −52% and from −8% to −82%, respectively, for Tomato. On the other hand, annual yield increase from −16% to 35%, from −5% to 17%, and from −36% to 59% was observed with the application of compost1, compost1+biochar, and compost2, respectively, for Onion, and from 9% to 37%, from 56% to 100%, and from 4% to 22% with the application of compost1, compost1+biochar, and compost2, respectively, for Rape (Figure 2). Biochar and compost2+biochar combination, however, has reduced the yield of these crops (Figure 2). Moreover, biochar alone increased yields of all crops in the first year, especially Aubergine. In contrast, a decrease in yield was observed in the second year, except for Aubergine and Lettuce (Figure 2).

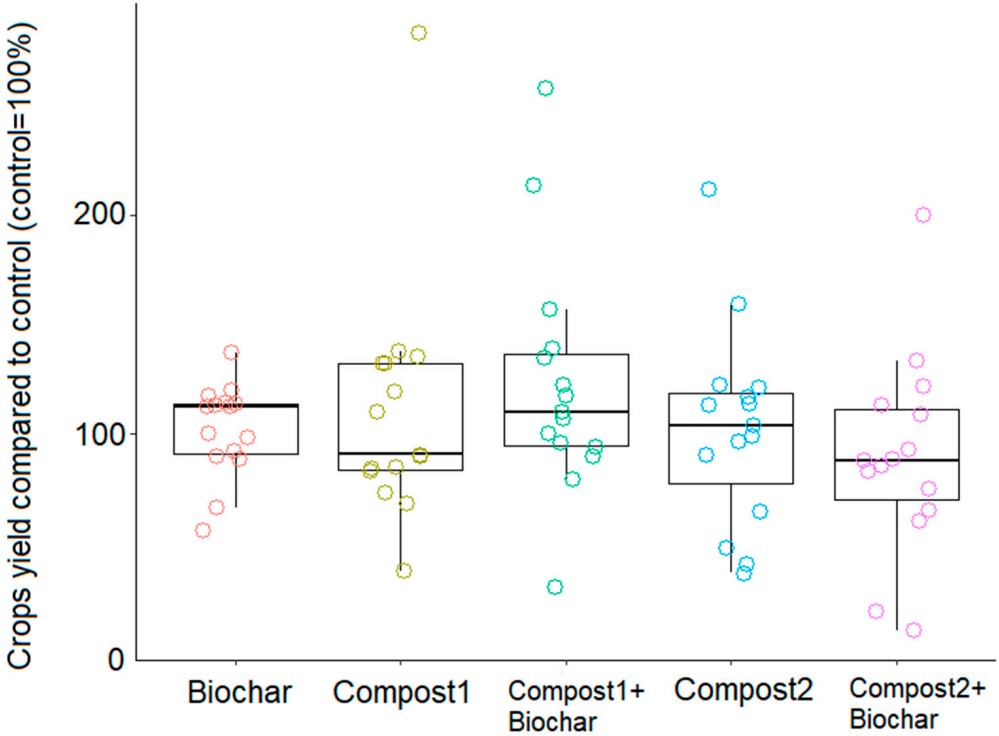

**Figure 1.** Overall crops yield compared to the control (100%) for each of the five applied treatments during two years of growth. The lower and upper bounds of the boxplots show the first and third quartiles (the 25th and 75th percentiles), the middle line shows the median, whiskers above and below the boxplot indicate inter-quartile range, and points beyond the whiskers indicate outlying points.

**Table 3.** Soil chemical parameters after one and two years of organic amendment application. Different letters within each column indicate significantly different groups (Duncan test, $p < 0.05$).

| Treatment | Year | pH | FDA | EC | C org. | $NH_4$ | Total N | $P_2O_5$ |
|---|---|---|---|---|---|---|---|---|
| Control | I | 7.53 a | 0.53 a | 247.68 a | 0.25 a | 0.19 a | 16.8 a | 96.65 a |
| Biochar | I | 7.57 a | 0.48 a | 216.50 a | 0.23 a | 0.30 b | 30.9 b | 126.24 b |
| Compost1 | I | 7.47 a | 0.59 a | 203.48 a | 0.24 a | 0.35 b | 35.9 b | 119.64 b |
| Compost2 | I | 7.48 a | 0.55 a | 193.80 a | 0.29 ab | 0.57 c | 47.9 c | 130.81 b |
| Compost1+Biochar | I | 7.58 a | 0.63 b | 215.75 a | 0.31 ab | 1.19 d | 67.5 d | 132.88 b |
| Compost2+Biochar | I | 7.54 a | 0.54 a | 343.10 b | 0.32 ab | 0.42 bc | 48.2 c | 119.24 b |
| Control | II | 5.82 b | 0.21 c | 217.98 a | 0.27 a | 0.19 a | 30.2 b | 85.34 a |
| Biochar | II | 5.70 b | 0.21 c | 192.77 a | 0.37 b | 0.31 b | 90.2 e | 110.41 b |
| Compost1 | II | 5.87 b | 0.34 c | 267.5 ab | 0.32 ab | 0.66 c | 41.7 c | 105.76 b |
| Compost2 | II | 6.00 b | 0.23 c | 216.88 a | 0.34 ab | 0.67 c | 36.8 b | 124.9 b |
| Compost1+Biochar | II | 5.99 b | 0.24 c | 201.05 a | 0.32 ab | 1.02 d | 23.8 a | 114.94 b |
| Compost2+Biochar | II | 5.82 b | 0.12 d | 216.15 a | 0.37 b | 0.41 bc | 23.6 a | 113.62 b |

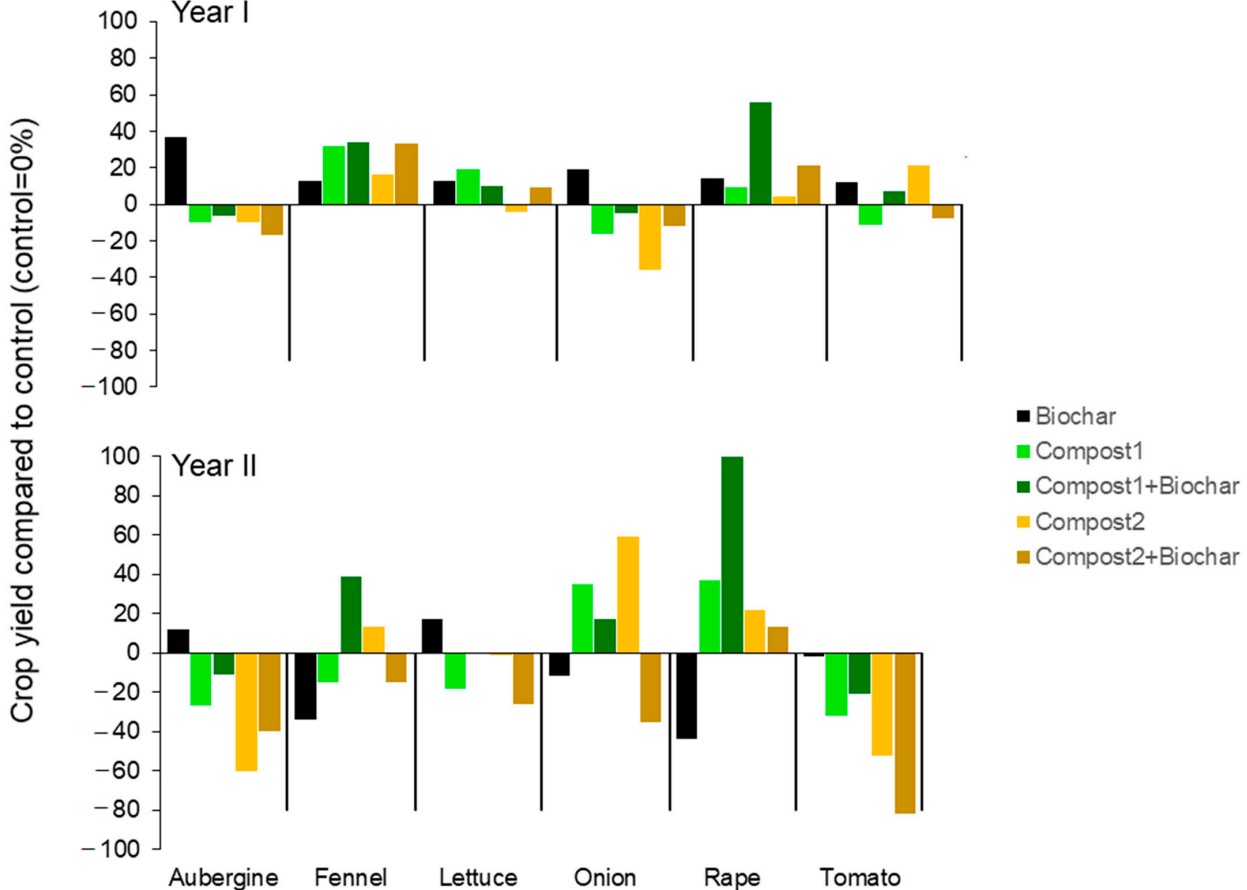

**Figure 2.** Specific response of each of the six crops, i.e., Aubergine (*Solanum melongena*), Fennel (*Foeniculum vulgare*), Lettuce (*Lactuca sativa*), Onion (*Allium cepa*), Rape (*Cucurbita pepo*), and Tomato (*Lycopersicon esculentum*), used during the experiment while applying each of the five different treatments. Specific response is represented by the crops yield comparing to the control (0%).

### 3.4. Soil Chemistry Drives Vegetable Yields

The effect of organic amendments either alone or in a mixture, on plant growth was largely influenced by soil chemistry (Figure 3). In detail, growth of plants treated with biochar alone was mainly affected by soil $NH_4$ content in the first year, while total nitrogen content was the main factor affecting growth in the second year. Compost1 regulated plant growth in the first year by changing pH and $P_2O_5$ content, and in the second year by total

nitrogen content and FDA content (Figure 3). In contrast, the mixture of compost1 and biochar were regulated by $P_2O_5$ content in the first year and by total nitrogen, $NH_4$, $P_2O_5$, and FDA in the second year. For compost2, growth was mainly controlled by total nitrogen content and EC in the first year, while $NH_4$ and FDA were more present in the second year. Nevertheless, the arrangement of the compost2+biochar mixture was determined by total soil nitrogen content in the first year, while pH and organic carbon played a role in the second year. The control, on the other hand, was determined by pH, $NH_4$ and organic carbon in the first year and total nitrogen content in the second year (Figure 3).

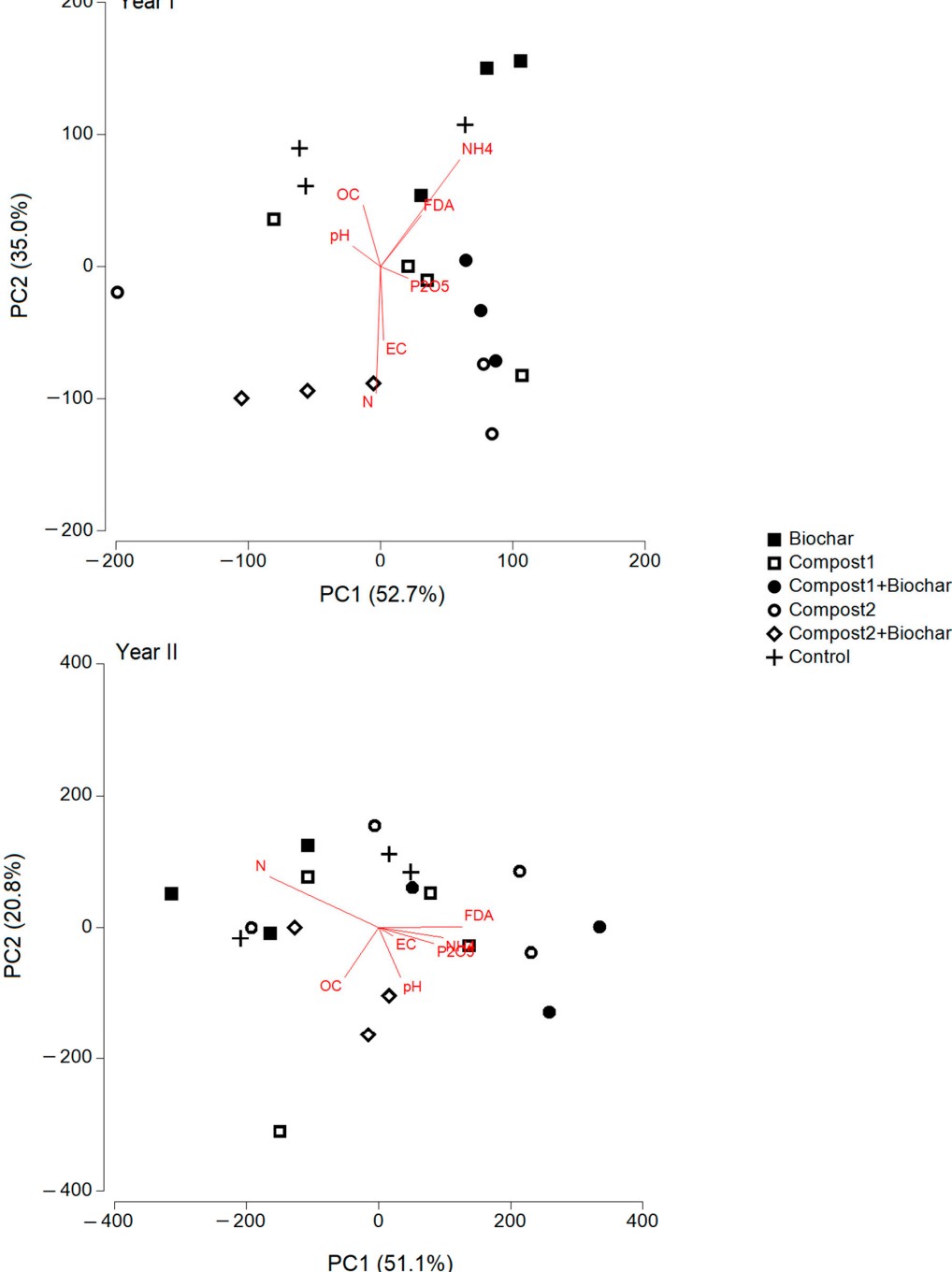

**Figure 3.** Principal component analysis (PCA) between biochar, compost 1, compost 2, compost1+Biochar, Compost2+Biocar and Control, based on chemical and biochemical parameters.

## 4. Discussion

Our experiment was conducted in an experimental field and greenhouse situated in the Napoli district (southern Italy) on the slopes of Mount Vesuvius. The results showed a great variability of crop yield depending on the crop species, applied treatments and the experimental year. In general, biochar showed a growth-promoting effect in the first year, ranging from 12% in Tomato to 37% in Aubergine, compared to the control. However, the positive effect of biochar persisted only in Aubergine and Lettuce, while it proved negative in the other crops, reaching −44% in Rape. On the other hand, the other treatments showed different responses depending on the crop species and the experimental year. In general, after two years, the mixture of compost1 and biochar proved to be the best treatment for the growth of Fennel and Rape with an increase of 39% and 100%, respectively, while in Onion, compost2 alone showed the highest growth of 59% compared to the other treatments. In Tomato, Aubergine and Lettuce, all other treatments showed negative growth responses compared to the control during 2nd year. Accordingly, Trupiano et al. [31] showed that addition of biochar and compost to a nutrient-poor soil had a positive effect on lettuce plant growth, but no positive synergistic or summative effects of compost and biochar were observed in combination. Accordingly, in a glasshouse experiment at the University of Toronto, Seehausen et al. [32] showed that the mixture of biochar and compost had generally neutral or antagonistic interactive effects on plant growth of both annual *Abutilon theophrasti* and perennial *Salix purpurea*. However, our results suggest that the combination of biochar and compost could have either positive (e.g., compost1+biochar for Fennel and Rape growth) or negative (e.g., compost2+biochar for Fennel, Lettuce, Onion, and Rape growth) effects depending on the plant species and compost types.

Our results also showed significant differences in the chemical properties of the organic matters used in terms of elemental content and molecular compositions. In general, biochar had an exceptionally high C/N ratio, followed by compost1, while it was dramatically low for compost2. The C/N ratio is linked to N mineralization because it is stoichiometrically related to the demand of saprophytic microbes [33,34]. Organic C or N can limit microbial proliferation when the C/N ratio is below or above the threshold of 30. In other words, if the C/N ratio exceeds 30, microbial decomposers of organic matter will not release inorganic N and could meet their N requirements from the soil inorganic N pool known as N immobilization, reducing the availability of this pool to plants. On the other hand, if the C/N ratio is below 30, then these microorganisms release excess N, often as NH4, and contribute to the soil inorganic N pool through mineralization [35]. Therefore, the negative effects of biochar with an extremely high C/N ratio on Fennel, Onion, Rape, and Tomato could be due to N immobilization. To efficiently utilize soil organic matter, microbes require both organic C and N at a relatively constant stoichiometric ratio [22], which could explain the large variability in our results. The inhibitory effects on plant growth could also be partially explained by the organic C quality of the compost. Our results showed that the O-alkyl-C fraction associated with polysaccharides and sugars was particularly high in both composts and very low in biochar, and that the di-O-alkyl-C fraction was equally low in both biochar and compost 2 and relatively higher in compost1. There is evidence that N mineralization is negatively correlated with simple sugars, other carbohydrates, and cellulose [34]. Since sugars and carbohydrates are readily available for microbial consumption, the rapidly growing microbial biomass will consume the available soil nutrients to meet their nutritional requirement, leading to rapid N immobilization, which would explain the inhibitory effects of both composts, even if they have low C/N ratios. Conversely, the positive effects on plant growth could be partly explained by the high content of methoxyl-C and carboxyl-C NMR regions and a high N content in both composts, especially in compost2. Indeed, both alkyl C and methoxyl C fractions showed high peaks in compost 2, followed by compost 1. Similarly, Bonanomi et al. [22] found that lettuce growth was promoted by non-pyrogenic organic amendments with relatively high content of methoxyl-C and carboxyl C and high N content, confirming the known positive effect of mineral and organic N sources on plant development [36].

It has been suggested that organic amendments, particularly biochar, have potential for soil improvement due to their unique physical, chemical, and biological properties [37]. Our results show that soil chemical compositions varied significantly between the first and second year of treatments. The addition of biochar alone or in combination with compost significantly increased soil organic carbon content over two years. However, soil carbon is widely recognized as a key indicator of soil fertility [38]. In addition, significant increases in pH and EC were observed in all compost and biochar treatments compared to the control, in the 1st year. The significant increases in soil pH are likely due to the mineral ash content of the various organic materials and result in a liming effect that increases soil pH [6]. In contrast, the observed decrease in soil pH, an acidification of the soil in the 2nd year in all treatments, could explain the decrease in crop yields, as pH is a crucial factor affecting microbial activity. In agreement with our results, it was reported that the addition of biochar leads to a decrease in pH due to the adsorption of $NH_3$, thus preventing the dissolution of $NH_3$ in the compost solution to release OH-ions [39]. Otherwise, it was observed that the stimulation of crop yield was higher in soils with an initial lower pH as often occur in tropical soils [40]. We also observed a significant increase in soil N, P, and $NH_4$ because of the organic amendments. This increase in soil nutrient content could be primarily related to the nutrient content of the amendments, which of course depends on the source and type of organic matter [41] and, in the case of biochar, feedstock, pyrolysis process, and pyrolysis temperature [42]. Therefore, the change in soil pH due to the organic amendments may have increased the available nutrient concentrations in the soil.

## 5. Conclusions

Our study suggests that the biochar-compost mixture in a Mediterranean climate has very different effects on yield in relation to crop types and amendments type. The factors contributing to this variability could depend on the significant differences in chemical compositions, in terms of elemental content and molecular compositions of biochar and the two composts. The chemical composition of the soil also varied significantly between the first and second years in terms of pH and enzymatic activity (FDA). These factors could explain why crop yields showed large variations in the two treatment years. In addition, yield was significantly affected by crop type and the interaction between crop type and experimental year. Based on these results, future research is needed to investigate the mechanisms underlying the chemical and microbiological nature of the synergistic and antagonistic interactions between biochar and organics so that practical guidelines can be developed to ensure reliable performance of biochar blends on crop productivity in different agricultural systems.

**Author Contributions:** Conceptualization, M.I., F.V. and G.B.; methodology, G.B., G.I. and F.V.; software, M.I.; validation, G.B., F.V. and M.I.; formal analysis, T.C.S.; investigation, G.I., A.S. and F.I.; resources, G.B. and F.V.; data curation, M.I.; writing—original draft preparation, M.I. and G.B.; writing—review and editing, G.B. and M.I.; visualization, A.S.; supervision, G.B. and M.I.; project administration, G.B.; funding acquisition, F.V. All authors have read and agreed to the published version of the manuscript.

**Funding:** This study was supported by MISE CRESO grant number protection no. F/050421/01-03/X32; Viabio no. F/200095/01-03/X45; Regione Campania (Project Dioniso, no. B98H19005010009).

**Institutional Review Board Statement:** Not applicable.

**Informed Consent Statement:** Not applicable.

**Data Availability Statement:** Not applicable.

**Conflicts of Interest:** The authors declare no conflict of interest.

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
