# Peer review of "Biochar and Compost Application either Alone or in Combination Affects Vegetable Yield in a Volcanic Mediterranean Soil"

_agronomy, doi:10.3390/agronomy12091996_

Round 1

Reviewer 1 Report

The manuscript titled “ Biochar and compost application either alone or in combination affect vegetable yield in a volcanic Mediterranean soil“. I find the idea interesting and in line with the aim of the journal. I have some concerns about the experimental setup to justify what the authors claim. Moreover, the rationale behind some of the data presented was not entirely clear. I also recommend to the authors improve their references by conducting a more extensive review of international literature. Particularly, the introduction statements are not supported by the references selected by the authors. The logic of some sentences is also questionable. Below is my point-to-point analysis of the manuscript.

 Abstract

The abstract is not properly written, it should be crisp, it should contain an introduction aim hypothesis aim result, and conclusion. The introduction section is too long in the abstract; one line of the background of the study in the abstract attracts the reader the most. A connective link is missing between different sections. Also, the concluding part of the introduction is missing at the end of the introduction. The author should make the introduction section crisp and to the point related to research, which I don't find in the present form of the manuscript. 

My main concern with a manuscript is the statical test.
what is the value of n while calculating ANOVA? n Value (3) used in the manuscript is too few to examine the normal distribution of variables in the sample, however, the Shapiro-Wilk test is appropriate for samples from 3 to 5000 but for the lesser value of n it receives the non-normal distribution. Thus ANOVA which is a parametrical test is incorrect for such small samples.

 Error Bar

Secondly, the error bar in the figure are not properly explained it seems they correspond to standard deviation which does not make any sense, that is merely for decorative purposes. I highly recommend using a 95% confidence interval instead of a Standard deviation, in the error bar.

PCA

PCA Graph is not properly explained.  For example, characteristics correlated to different attribute is not explained; the Length of different vector (Variability of Variable) is not explained, etc. similarly other properties of the PCA graph must be explained.

 Data Distribution

Author should mention the data set that passes the normality test. The author has applied parametric and nonparametric tests, he has not explained the reason of the non-parametric test is it because of nonnormal distribution or because of variance.

Although the study is interesting and could be useful for a certain group of scientific fraternity, therefore, I would suggest improving the manuscript substantially, giving it a chance for the next round, because the subject is interesting. However, even an interesting subject depends on statics.

Author Response

Reviewer #1

The manuscript titled “Biochar and compost application either alone or in combination affect vegetable yield in a volcanic Mediterranean soil“. I find the idea interesting and in line with the aim of the journal. I have some concerns about the experimental setup to justify what the authors claim. Moreover, the rationale behind some of the data presented was not entirely clear. I also recommend to the authors improve their references by conducting a more extensive review of international literature. Particularly, the introduction statements are not supported by the references selected by the authors. The logic of some sentences is also questionable. Below is my point-to-point analysis of the manuscript.

Response: We would like to thank the reviewer for his valuable comments and also for his concerns. In order to improve our manuscript to make it more suitable for the journal, we have taken the reviewer's comments seriously and have tried to address each one. Please find bellow a more detailed description of our responses to the concerns raised, and also in the track version manuscript.

Abstract

The abstract is not properly written, it should be crisp, it should contain an introduction aim hypothesis aim result, and conclusion. The introduction section is too long in the abstract; one line of the background of the study in the abstract attracts the reader the most. A connective link is missing between different sections. Also, the concluding part of the introduction is missing at the end of the introduction. The author should make the introduction section crisp and to the point related to research, which I don't find in the present form of the manuscript.

Response: We agree with the reviewer on his points and we have changed the abstract accordingly.

The introduction section of the abstract was changed as follows: “The aim of this work was to compare the application of biochar, compost, and their mixtures on soil fertility and crop yields using a volcanic Mediterranean soil.” Lines 19-20.

Moreover, we added a link between each section of the abstract as suggested by the reviewer.

The M&M section of the abstract was adjusted as follows: “When selected, the OAs were characterized chemically for organic carbon (OC), total N, pH, electric conductivity (EC) and the bulk fraction of organic matter using 13C CPMAS NMR spectroscopy.” Lines 23-35.

Please check the abstract in the track-version manuscript for more details.

My main concern with a manuscript is the statical test.

what is the value of n while calculating ANOVA? n Value (3) used in the manuscript is too few to examine the normal distribution of variables in the sample, however, the Shapiro-Wilk test is appropriate for samples from 3 to 5000 but for the lesser value of n it receives the non-normal distribution. Thus ANOVA which is a parametrical test is incorrect for such small samples.

Response: We agree with the reviewer that the p-value is highly sensitive to the number of replicates, i.e., size of the sample, the larger the sample the more likely a difference to be detected. However, we would like to inform the reviewer that the p-value is more influenced by the effect size and the spread of data. In our case, we have 6 different treatments on 6 different crops with three replicates each, totalling 108 observations, which would reduce the impact of random error. Moreover, the magnitude of differences between groups also plays a role. If there is a large magnitude of difference then it will be easier to detect differences between groups. If there are two studies with equal sample sizes, which are free of error measuring two different relationships, the relationship with the larger magnitude of effect (e.g., difference between groups) will have a small P value as compared to the study with the smaller magnitude of effect. In our case, the study is of long term taking 2 years, the standard deviation between treatments on crops in the first year showed high differences compared to the second year, increasing thus the magnitude of the effect. Finally, we would like to inform the reviewer that the minimum sample size requirement in the literature to run an ANOVA test successfully is 3.

Error Bar

Secondly, the error bar in the figure are not properly explained it seems they correspond to standard deviation which does not make any sense, that is merely for decorative purposes. I highly recommend using a 95% confidence interval instead of a Standard deviation, in the error bar.

Response: We would like to inform the reviewer that we did not understand his comment regarding the error bar, first because the reviewer did not precise which figure he is talking about, and second because in our manuscript we have no figure containing the error bars. Probably the reviewer would be referring to Figure 1 with the boxplot, however, the whiskers of the boxplot are the two lines outside the box, that go from the minimum to the lower quartile (the start of the box) and then from the upper quartile (the end of the box) to the maximum. The length of the upper whisker is the largest value that is no greater than the third quartile plus 1.5 times the interquartile range. Moreover, in our statistical analyses, we used 95% confidence interval.

PCA

PCA Graph is not properly explained.  For example, characteristics correlated to different attribute is not explained; the Length of different vector (Variability of Variable) is not explained, etc. similarly other properties of the PCA graph must be explained.

Response: Done. A small paragraph was added to the material and methods section in the data analysis part regarding the PCA as follows: “Moreover, Principal Component Analysis (PCA), a multivariate approach, was per-formed to evaluate the treatments variability based on chemical and biochemical parameters. The length of each eigenvector is proportional to the variance in the data for that element. The angle between eigenvectors represents correlations among different elements.” Lines 191-195

Data Distribution

Author should mention the data set that passes the normality test. The author has applied parametric and nonparametric tests, he has not explained the reason of the non-parametric test is it because of nonnormal distribution or because of variance.

Response: We would like to inform the reviewer that in our study case we have applied only parametric statistical test, i.e., ANOVA test, because we know that our data is normally disturbed since the majority of data points are relatively similar, meaning they occur within a small range of values with fewer outliers on the high and low ends of the data range. On the other hand, a non-parametric test does not assume anything about the underlying distribution of data. Moreover, nonparametric tests can perform well with non-normal continuous data if you have a sufficiently large sample size, which is not our case since we have a limited sample size of 3 replicates.

Although the study is interesting and could be useful for a certain group of scientific fraternity, therefore, I would suggest improving the manuscript substantially, giving it a chance for the next round, because the subject is interesting. However, even an interesting subject depends on statics.

Response: We agree with the reviewer and we would like to show our gratitude again for his efforts to evaluate our manuscript. Indeed, statistical analyses are the most important in order to extract significance level out of the data. Since we have only three independent variables, i.e., crop type, treatment, and year, we have chosen to apply simple ANOVA test in order to calculate the variability among them. Following the reviewer’s recommendations, we have provided answers to the raised concerns and changed the manuscript accordingly.

Reviewer 2 Report

Introduction:

1.     Line 62: The most commonly used biochar is wood based? I have seen many types of biochar are not wood based.

2.     Line 91: add citation here.

M&M:

1.     What’s the biochar price in Italy now?

2.     Do you have weather data for the two years of your study?

Reviewer 3 Report

22,29 – The authors write the names of the samples with a lowercase letter, another time with a capital letter. Please change it.

23: „550 °C. – Remove the space. Apply this throughout the article.

98,99, 228,229 : „CaCO3, P2O5,” – please use subscript.

124-127: „Three types of organic amendments were selected: (1) compost1, from olive mill waste and orchard pruning residues; (2) compost2, from olive mill waste, animal manure, and wool residues; (3) biochar, produced from beech wood (Fagus sylvatica), pyrolyzed at 550°C.” – Please describe the conditions under which these fertilizers were created. This is crucial.

149: „m2” – Superscript

154: „20 t ha-1” – I suggest writing Mg‧ha-1

233,260 – Please change the table style.

Check the Paper with Guide for Authors.

Author Response

Reviewer #3

22,29 – The authors write the names of the samples with a lowercase letter, another time with a capital letter. Please change it.

Done. The names were changed as suggested.

23: „550 °C. – Remove the space. Apply this throughout the article.

Done. The space was removed all along the article as requested. Please check the track version manuscript for more details.

98,99, 228,229 : „CaCO3, P2O5,” – please use subscript.

Done. Subscripts were correctly used throughout the manuscript as recommended.

124-127: „Three types of organic amendments were selected: (1) compost1, from olive mill waste and orchard pruning residues; (2) compost2, from olive mill waste, animal manure, and wool residues; (3) biochar, produced from beech wood (Fagus sylvatica), pyrolyzed at 550°C.” – Please describe the conditions under which these fertilizers were created. This is crucial.

We agree with the reviewer and the following paragraph was added: “Composting mixtures were prepared by mixing either dried or fresh olive mill waste with different organic materials used as bulking agents and N-sources, such as animal manure, orchard pruning residues, and wool residues. In a pilot plant, composts were prepared using the turning-pile system in trapezoidal piles (2 m long, 1 m wide at the base, and 1 m high). The piles were turned every two weeks during the bio-oxidative phase (approximately 16-20 weeks, from June to October). The mixtures were matured over a two-month period. Temperature and moisture were used as monitoring parameters to track the progress of composting. Water was added during turning to maintain moisture content in the range of 40-60%. On the other hand, biochar of beech wood (Fagus sylvatica) was made by slow pyrolysis at 550 °C, and was used as stable carbon material, to improve soil physical properties.” Lines 127-137.

149: „m2” – Superscript

Done.

154: „20 t ha-1” – I suggest writing Mg‧ha-1

Done. It was changed as suggested.

233,260 – Please change the table style. Check the Paper with Guide for Authors.

Done. The style of the tables was unified as suggested by the reviewer. We would like to take the opportunity to thank the reviewer for his positive comments that allowed us to improve our manuscript.

Round 2

Reviewer 1 Report

I agree with the author and accept all the changes. My recommendation is to accept the manuscript as it is in its present form.